# FMix: Enhancing Mixed Sample Data Augmentation

## Abstract

Mixed Sample Data Augmentation (MSDA) has received increasing attention in recent years, with many successful variants such as MixUp and CutMix. We analyse MSDA from an information theoretic perspective, characterising learned models in terms of how they impact the models' perception of the data. Ultimately, our analyses allow us to decouple two complementary properties of augmentations that are useful for reasoning about MSDA. From insight on the efficacy of CutMix in particular, we subsequently propose FMix, an MSDA that uses binary masks obtained by applying a threshold to low frequency images sampled from Fourier space. FMix improves performance over MixUp and CutMix for a number of models across a range of data sets and problem settings, obtaining new state-of-the-art results on CIFAR-10 and Fashion-MNIST.

## 1 Introduction

Recently, a plethora of approaches to Mixed Sample Data Augmentation (MSDA) have been proposed which obtain state-of-the-art results, particularly in classification tasks (Chawla et al., 2002; Zhang et al., 2017; Tokozume et al., 2017; 2018; Inoue, 2018; Yun et al., 2019; Takahashi et al., 2019; Summers and Dinneen, 2019). MSDA involves combining data samples according to some policy to create an augmented data set on which to train the model. The policies so far proposed can be broadly categorised as either combining samples with interpolation (e.g. MixUp) or masking (e.g. CutMix). Traditionally, augmentation is viewed through the framework of statistical learning as Vicinal Risk Minimisation (VRM) (Vapnik, 1999; Chapelle et al., 2001). Given some notion of the vicinity of a data point, VRM trains with vicinal samples in addition to the data points themselves. This is the motivation for MixUp (Zhang et al., 2017); to provide a new notion of vicinity based on mixing data samples. In the classical theory, validity of this technique relies on the strong assumption that the vicinal distribution precisely matches the true distribution of the data. As a result, the classical goal of augmentation is to maximally increase the data space, without changing the data distribution. Clearly, for all but the most simple augmentation strategies, the data distribution is in some way distorted. Furthermore, there may be practical implications to correcting this, as is demonstrated in Touvron et al. (2019). In light of this, three important questions arise regarding MSDA: What is good measure of the similarity between the augmented and the original data? Why is MixUp so effective when the augmented data looks so different? If the data is distorted, what impact does this have on trained models?

To construct a good measure of similarity, we note that the data only need be 'perceived' similar by the model. As such, we measure the mutual information between representations learned from the real and augmented data, thus characterising how well learning from the augmented data simulates learning from the real data. This measure clearly shows the data-level distortion of MixUp by demonstrating that learned representations are compressed in comparison to those learned from the un-augmented data. To address the efficacy of MixUp, we look to the information bottleneck theory of deep learning (Tishby and Zaslavsky, 2015). This theory uses the data processing inequality, summarised as 'post-processing cannot increase information', to suggest that deep networks progressively discard information about the input whilst preserving information about the targets. Through this lens, we posit that the distortion and subsequent compression induced by MixUp promotes generalisation by preventing the network from learning about highly sample-specific features in the data. Regarding the impact on trained models, and again armed with the knowledge that MixUp distorts learned functions, we show that MixUp acts as a kind of adversarial training (Good-

fellow et al., 2014), promoting robustness to additive noise. This accords with the theoretical result of Perrault-Archambault et al. (2020) and the robustness results of Zhang et al. (2017). However, we further show that MSDA does not generally improve adversarial robustness when measured as a worst case accuracy following multiple attacks as suggested by Carlini et al. (2019). In contrast to our findings regarding MixUp, we show that CutMix causes learned models to retain a good knowledge of the real data, which we argue derives from the fact that individual features extracted by a convolutional model generally only derive from one of the mixed data points. At the same time Cut-Mix limits the ability of the model to over-fit by dramatically increasing the number of observable data points, in keeping with the original intent of VRM. We go on to argue that by restricting to only masking a square region, CutMix imposes an unnecessary limitation. Indeed, it should be possible to construct an MSDA which uses masking similar to CutMix whilst increasing the data space much more dramatically. Motivated by this, we introduce FMix, a masking MSDA that uses binary masks obtained by applying a threshold to low frequency images sampled from Fourier space. Using our mutual information measure, we show that learning with FMix simulates learning from the real data even better than CutMix. We subsequently demonstrate performance of FMix for a range of models and tasks against a series of augmented baselines and other MSDA approaches. FMix obtains a new state-of-the-art performance on CIFAR-10 (Krizhevsky et al., 2009) without external data and Fashion MNIST (Xiao et al., 2017) and improves the performance of several state-of-the-art models (ResNet, SE-ResNeXt, DenseNet, WideResNet, PyramidNet, LSTM, and Bert) on a range of problems and modalities.

In light of our analyses, and supported by our experimental results, we go on to suggest that the compressing qualities of MixUp are most desirable when data is limited and learning from individual examples is easier. In contrast, masking MSDAs such as FMix are most valuable when data is abundant. We finally suggest that there is no reason to see the desirable properties of masking and interpolation as mutually exclusive. In light of these observations, we plot the performance of MixUp, FMix, a baseline, and a hybrid policy where we alternate between batches of MixUp and FMix, as the number of CIFAR-10 training examples is reduced. This experiment confirms our above suggestions and shows that the hybrid policy can outperform both MixUp and FMix.

## 2  RELATED WORK

In this section, we review the fundamentals of MSDA. Let $p_X(x)$ denote the input data distribution. In general, we can define MSDA for a given mixing function, $\mathrm{mix}(X_1, X_2, \Lambda)$, where $X_1$ and $X_2$ are independent random variables on the data domain and $\Lambda$ is the mixing coefficient. Synthetic minority over-sampling (Chawla et al., 2002), a predecessor to modern MSDA approaches, can be seen as a special case of the above where $X_1$ and $X_2$ are dependent, jointly sampled as nearest neighbours in feature space. These synthetic samples are drawn only from the minority class to be used in conjunction with the original data, addressing the problem of imbalanced data. The mixing function is linear interpolation, $\mathrm{mix}(x_1, x_2, \lambda) = \lambda x_1 + (1 - \lambda) x_2$, and $p_\Lambda = \mathcal{U}(0, 1)$. More recently, Zhang et al. (2017), Tokozume et al. (2017), Tokozume et al. (2018) and Inoue (2018) concurrently proposed using this formulation (as MixUp, Between-Class (BC) learning, BC+ and sample pairing respectively) on the whole data set, although the choice of distribution for the mixing coefficients varies for each approach. We refer to this as interpolative MSDA, where, following Zhang et al. (2017), we use the symmetric Beta distribution, that is $p_\Lambda = \mathrm{Beta}(\alpha, \alpha)$.

Recent variants adopt a binary masking approach (Yun et al., 2019; Summers and Dinneen, 2019; Takahashi et al., 2019). Let $M = \mathrm{mask}(\Lambda)$ be a random variable with $\mathrm{mask}(\lambda) \in \{0,1\}^n$ and $\mu(\mathrm{mask}(\lambda)) = \lambda$, that is, generated masks are binary with average value equal to the mixing coefficient. The mask mixing function is $\mathrm{mix}(\mathbf{x}_1, \mathbf{x}_2, \mathbf{m}) = \mathbf{m} \odot \mathbf{x}_1 + (1 - \mathbf{m}) \odot \mathbf{x}_2$, where $\odot$ denotes point-wise multiplication. A notable masking MSDA which motivates our approach is CutMix (Yun et al., 2019). CutMix is designed for two dimensional data, with $\mathrm{mask}(\lambda) \in \{0,1\}^{w \times h}$, and uses $\mathrm{mask}(\lambda) = \mathrm{rand\_rect}(w\sqrt{1-\lambda}, h\sqrt{1-\lambda})$, where $\mathrm{rand\_rect}(r_w, r_h) \in \{0,1\}^{w \times h}$ yields a binary mask with a shaded rectangular region of size $r_w \times r_h$ at a uniform random coordinate. CutMix improves upon the performance of MixUp on a range of experiments. For the remainder of the paper we focus on the development of a better input mixing function. Appendix A provides a discussion of the importance of the mixing ratio of the labels. For the typical case of classification with a cross entropy loss, the objective function is simply the interpolation between the cross entropy against each of the ground truth targets.

Table 1: Mutual information between VAE latent spaces ($Z_A$) and the CIFAR-10 test set ($I(Z_A; X)$), and the CIFAR-10 test set as reconstructed by a baseline VAE ($I(Z_A; \hat{X})$) for VAEs trained with a range of MSDAs. MixUp prevents the model from learning about specific features in the data. Uncertainty estimates are the standard deviation following 5 trials.

|          | $I(Z_A; X)$       | $I(Z_A; \hat{X})$ | MSE              |
|----------|-------------------|-------------------|------------------|
| Baseline | $78.05_{\pm 0.53}$ | $74.40_{\pm 0.45}$ | $0.256_{\pm 0.002}$ |
| MixUp    | $70.38_{\pm 0.90}$ | $68.58_{\pm 1.12}$ | $0.288_{\pm 0.003}$ |
| CutMix   | $83.17_{\pm 0.72}$ | $79.46_{\pm 0.75}$ | $0.254_{\pm 0.003}$ |

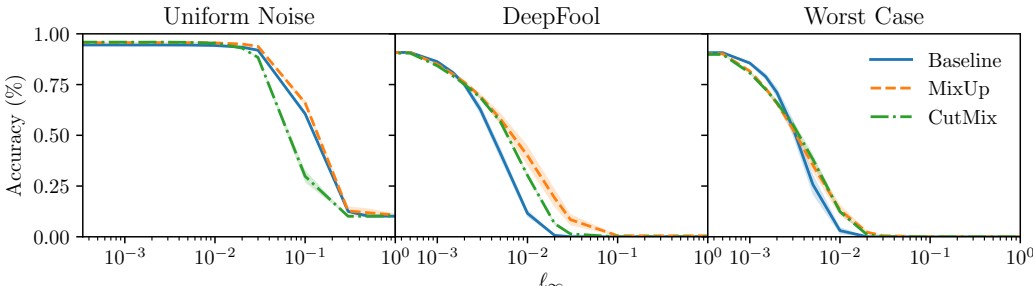

Figure 1: Robustness of PreAct-ResNet18 models trained on CIFAR-10 with standard augmentations (Baseline) and the addition of MixUp and CutMix to uniform noise, DeepFool (Moosavi-Dezfooli et al., 2016), and the worst case performance after multiple attacks including the additive uniform noise and DeepFool. MixUp improves robustness to adversarial examples with similar properties to images generated with MixUp (acting as adversarial training), but MSDA does not improve robustness in general. Shaded region indicates the standard deviation following 5 repeats.

## 3  ANALYSIS

We now analyse both interpolative and masking MSDAs with a view to distinguishing their impact on learned representations. We summarise previous analyses and theories (Zhang et al., 2017; Liang et al., 2018; Guo et al., 2019; He et al., 2019; Verma et al., 2019; Yun et al., 2019) in Appendix G. For our analysis, we desire a measure which captures the extent to which learning about the augmented data simulates learning about the original data. To achieve this, we propose training unsupervised models on real data and augmented data and then measuring the mutual information, the reduction in uncertainty about one variable given knowledge of another, between the representations they learn. In particular, we propose using Variational Auto-Encoders (VAEs) (Kingma and Welling, 2013), which provide a rich depiction of the salient or compressible information in the data (Higgins et al.). Denoting the latent space of a VAE trained on the original data as $Z_X$ and on some candidate augmentation $A$ as $Z_A$, in Appendix B we show that we can obtain a tractable lower bound, $I(Z_A; X)$, and upper bound, $I(Z_A; \hat{X})$ where $\hat{X}$ is the original data as reconstructed by a baseline VAE, for the intractable quantity $I(Z_A; Z_X)$. Table 1 gives these quantities for MixUp, CutMix, and a baseline. The results show that MixUp consistently reduces the amount of information that is learned about the original data. In contrast, CutMix manages to induce greater mutual information with the data than is obtained from just training on the un-augmented data. Crucially, the results present concrete evidence that interpolative MSDA differs fundamentally from masking MSDA in how it impacts learned representations.

Having shown this is true for VAEs, we now wish to understand whether the finding also holds for trained classifiers. To this end, in Figure 4 in the appendix we visualise the decisions made by a classifier using Gradient-weighted Class Activation Maps (Grad-CAMs) (Selvaraju et al., 2017). Grad-CAM finds the regions in an image that contribute the most to the network's prediction by taking the derivative of the model's output with respect to the activation maps and weighting them according to their contribution. If MixUp prevents the network from learning about highly specific features in the data we would expect more of the early features to contribute to the network output.

Clearly, it is difficult to ascertain whether this is the case from the examples in the figure, although there is some indication that it may be true. To verify empirically we compute the average sum of Grad-CAM heatmaps over the CIFAR-10 test set for 5 repeats (independently trained PreAct-ResNet18 models). We obtain the following scores: baseline - $146_{\pm 5}$, MixUp - $162_{\pm 3}$, CutMix - $131_{\pm 6}$. It is clear that on average more of the early features contribute to the decisions made by MixUp trained models and that this result is consistent across independent runs.

Following on from these observations, it is now pertinent to ask whether these different representations learned from MixUp give rise to practical differences other than just improved generalisation. Since it is our assessment that models trained with MixUp have an altered 'perception' of the data distribution, we suggest an analysis based on adversarial attacks, which involve perturbing images outside of the perceived data distribution to alter the given classification. We perform fast gradient sign method, standard gradient descent, projected gradient descent, additive uniform noise, and DeepFool (Moosavi-Dezfooli et al., 2016) attacks over the whole CIFAR-10 test set on PreAct-ResNet18 models subject to $\ell_\infty$ constraints using the Foolbox library (Rauber et al., 2020; 2017). The plots for the additive uniform noise and DeepFool attacks, given in Figure 1, show that MixUp provides an improvement over CutMix and the augmented baseline in this setting. This is because MixUp acts as a form of adversarial training (Goodfellow et al., 2014), equipping the models with valid classifications for images of a similar nature to those generated by the additive noise and DeepFool attacks. In Figure 1, we additionally plot the worst case robustness following all attacks as suggested by Carlini et al. (2019). These results show that the adversarial training effect of MixUp is limited and does not correspond to a general increase in robustness. We provide an enhanced depiction of these results in Appendix C.

## 4 FMIX: IMPROVED MASKING

Our finding is that the masking MSDA approach works because it effectively preserves the data distribution in a way that interpolative MSDAs do not, particularly in the perceptual space of a Convolutional Neural Network (CNN). We suggest that this derives from the fact that each convolutional neuron at a particular spatial position generally encodes information from only one of the inputs at a time. This could also be viewed as local consistency in the sense that elements that are close to each other in space typically derive from the same data point. To the detriment of CutMix, it would be easy for a model to learn about the augmentation since perfectly horizontal and vertical artefacts are unlikely to be a salient feature of the data. We contend that a method which retains the masking nature of CutMix but increases the space of possible shapes (removing the bias towards horizontal and vertical edges) may be able to induce an even greater knowledge of the un-augmented data in trained models as measured by our mutual information analysis. This should in turn correspond with improved accuracy. If we can increase the number and complexity of masks then the space of novel features (that is, features which occur due to edges in the mask) would become significantly larger than the space of features native to the data. As a result, it is highly unlikely that a model would be able to 'fit' to this information. This leads to our core motivation: to construct a masking MSDA which maximises the space of edge shapes whilst preserving local consistency.

For local consistency, we require masks that are predominantly made up of a single shape or contiguous region. We might think of this as trying to minimise the number of times the binary mask transitions from '0' to '1' or vice-versa. For our approach, we begin by sampling a low frequency grey-scale mask from Fourier space which can then be converted to binary with a threshold. We will first detail our approach for obtaining the low frequency image before discussing our approach for choosing the threshold. Let $Z$ denote a complex random variable with values on the domain $\mathcal{Z} = \mathbb{C}^{w \times h}$, with density $p_{\Re(Z)} = \mathcal{N}(\mathbf{0}, \boldsymbol{I}_{w \times h})$ and $p_{\Im(Z)} = \mathcal{N}(\mathbf{0}, \boldsymbol{I}_{w \times h})$, where $\Re$ and $\Im$ return the real and imaginary parts of their input respectively. Let $\mathrm{freq}(w, h)\,[i, j]$ denote the magnitude of the sample frequency corresponding to the $i$, $j$'th bin of the $w \times h$ discrete Fourier transform. We can apply a low pass filter to $Z$ by decaying its high frequency components. Specifically, for a given decay power $\delta$, we use

$$\mathrm{filter}(\mathbf{z}, \delta)[i, j] = \frac{\mathbf{z}[i, j]}{\mathrm{freq}(w, h)\,[i, j]^{\delta}} \ . \tag{1}$$

Defining $\mathcal{F}^{-1}$ as the inverse discrete Fourier transform, we can obtain a grey-scale image with

$$G = \Re\big(\mathcal{F}^{-1}\big(\mathrm{filter}\,(Z, \delta)\big)\big) \ . \tag{2}$$

Mask

Image 1

Image 2

FMix

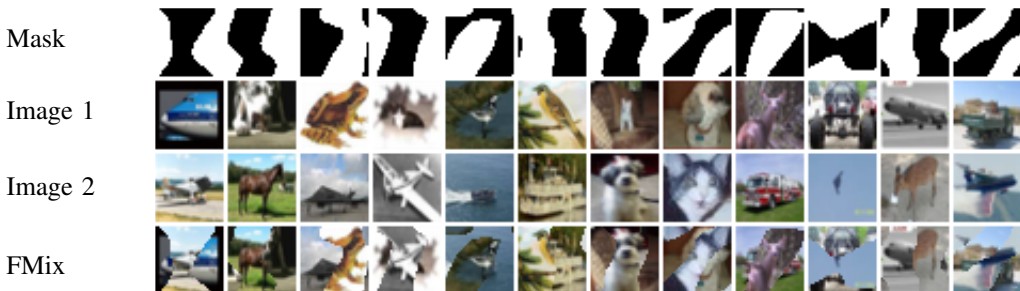

Figure 2: Example masks and mixed images from CIFAR-10 for FMix with $\delta = 3$ and $\lambda = 0.5$.

All that now remains is to convert the grey-scale image to a binary mask such that the mean value is some given $\lambda$. Let $\text{top}(n, \mathbf{x})$ return a set containing the top $n$ elements of the input $\mathbf{x}$. Setting the top $\lambda wh$ elements of some grey-scale image $\mathbf{g}$ to have value '1' and all others to have value '0' we obtain a binary mask with mean $\lambda$. Specifically, we have

$$\text{mask}(\lambda, \mathbf{g})[i, j] = \begin{cases} 1, & \text{if } \mathbf{g}[i, j] \in \text{top}(\lambda wh, \mathbf{g}) \\ 0, & \text{otherwise} \end{cases} . \tag{3}$$

To recap, we first sample a random complex tensor for which both the real and imaginary part are independent and Gaussian. We then scale each component according to its frequency via the parameter $\delta$ such that higher values of $\delta$ correspond to increased decay of high frequency information. Next, we perform an inverse Fourier transform on the complex tensor and take the real part to obtain a grey-scale image. Finally, we set the top proportion of the image to have value '1' and the rest to have value '0' to obtain our binary mask. Although we have only considered two dimensional data here it is generally possible to create masks with any number of dimensions. We provide some example two dimensional masks and mixed images (with $\delta = 3$ and $\lambda = 0.5$) in Figure 2. We can see that the space of artefacts is significantly increased, furthermore, FMix achieves $I(Z_A; X) = 83.67_{\pm 0.89}$, $I(Z_A; \hat{X}) = 80.28_{\pm 0.75}$, and MSE $= 0.255_{\pm 0.003}$, showing that learning from FMix simulates learning from the un-augmented data to an even greater extent than CutMix.

## 5 EXPERIMENTS

We now perform a series of experiments to compare the performance of FMix with that of MixUp, CutMix, and augmented baselines. For each problem setting and data set, we provide exposition on the results and any relevant caveats. Throughout, our approach has been to use the hyper-parameters which yield the best results in the literature for each setting. Unless otherwise stated, we use $\alpha = 1$ for the distribution of $\lambda$. For FMix, we use $\delta = 3$ since this was found to produce large artefacts with sufficient diversity. We perform an ablation of both parameters in Appendix H, reporting results for $5$ fold cross validation. We perform repeats where possible and report the average performance and standard deviation after the last epoch of training. A complete discussion of the experimental set-up can be found in Appendix E along with the standard augmentations used for all models on each data set. Additional experiments on point cloud and audio classification are given in Appendix D. In all tables, we give the best result and results that are within its margin of error in **bold**. We discuss any cases where the results obtained by us do not match the results obtained by the authors in the accompanying text, and give the authors results in parentheses. Uncertainty estimates are the standard deviation over $5$ repeats. Code for all experiments is given in the supplementary material.

**Image Classification** We first discuss image classification results on the CIFAR-10/100 (Krizhevsky et al., 2009), Fashion MNIST (Xiao et al., 2017), and Tiny-ImageNet (Stanford, 2015) data sets. We train: PreAct-ResNet18 (He et al., 2016), WideResNet-28-10 (Zagoruyko and Komodakis, 2016), DenseNet-BC-190 (Huang et al., 2017) and PyramidNet-272-200 (Han et al., 2017). For PyramidNet, we additionally apply Fast AutoAugment (Lim et al., 2019), a successor to AutoAugment (Cubuk et al., 2019a), and ShakeDrop (Yamada et al., 2018) following Lim et al. (2019). The results in Table 2 show that FMix offers a significant improvement over the other

Table 2: Image classification accuracy for our approach, FMix, against baselines for: PreAct-ResNet18 (ResNet), WideResNet-28-10 (WRN), DenseNet-BC-190 (Dense), PyramidNet-272-200 + ShakeDrop + Fast AutoAugment (Pyramid). Parentheses indicate author quoted result.

| Data set | Model | Baseline | FMix | MixUp | CutMix |
|---|---|---|---|---|---|
| CIFAR-10 | ResNet | $94.63_{\pm0.21}$ | $\mathbf{96.14}_{\pm0.10}$ | $95.66_{\pm0.11}$ | $96.00_{\pm0.07}$ |
| | WRN | $95.25_{\pm0.10}$ | $96.38_{\pm0.06}$ | $(\mathbf{97.3})\ \mathbf{96.60}_{\pm0.09}$ | $96.53_{\pm0.10}$ |
| | Dense | $96.26_{\pm0.08}$ | $\mathbf{97.30}_{\pm0.05}$ | $(\mathbf{97.3})\ 97.05_{\pm0.05}$ | $96.96_{\pm0.01}$ |
| | Pyramid | $98.31$ | $\mathbf{98.64}$ | $97.92$ | $98.24$ |
| CIFAR-100 | ResNet | $75.22_{\pm0.20}$ | $\mathbf{79.85}_{\pm0.27}$ | $(78.9)\ 77.44_{\pm0.50}$ | $79.51_{\pm0.38}$ |
| | WRN | $78.26_{\pm0.25}$ | $82.03_{\pm0.27}$ | $(\mathbf{82.5})\ 81.09_{\pm0.33}$ | $81.96_{\pm0.40}$ |
| | Dense | $81.73_{\pm0.30}$ | $\mathbf{83.95}_{\pm0.24}$ | $83.23_{\pm0.30}$ | $82.79_{\pm0.46}$ |
| Fashion | ResNet | $95.70_{\pm0.09}$ | $\mathbf{96.36}_{\pm0.03}$ | $96.28_{\pm0.08}$ | $96.03_{\pm0.10}$ |
| | WRN | $95.29_{\pm0.17}$ | $\mathbf{96.00}_{\pm0.11}$ | $95.75_{\pm0.09}$ | $95.64_{\pm0.20}$ |
| | Dense | $95.84_{\pm0.10}$ | $96.26_{\pm0.10}$ | $\mathbf{96.30}_{\pm0.04}$ | $96.12_{\pm0.13}$ |
| Tiny-ImageNet | ResNet | $55.94_{\pm0.28}$ | $61.43_{\pm0.37}$ | $55.96_{\pm0.41}$ | $\mathbf{64.08}_{\pm0.32}$ |

Table 3: Classification performance for a ResNet101 trained on ImageNet for 90 epochs with a batch size of 256, and evaluated on ImageNet and ImageNet-a, adversarial examples to ImageNet. Note that Zhang et al. (2017) (MixUp) use a batch size of 1024 and Yun et al. (2019) (CutMix) train for 300 epochs, so these results should not be directly compared.

| Data set | $\alpha$ | Baseline | | FMix | | MixUp | | CutMix | |
|---|---|---|---|---|---|---|---|---|---|
| | | Top-1 | Top-5 | Top-1 | Top-5 | Top-1 | Top-5 | Top-1 | Top-5 |
| ImageNet | 1.0 | 77.28 | 93.63 | $\mathbf{77.42}$ | $\mathbf{93.92}$ | 75.89 | 93.06 | 76.92 | 93.55 |
| | 0.2 | | | $\mathbf{77.70}$ | $\mathbf{93.97}$ | 77.23 | 93.81 | 76.72 | 93.46 |
| ImageNet-a | 1.0 | 4.08 | 28.87 | 7.19 | 33.65 | $\mathbf{8.69}$ | $\mathbf{34.89}$ | 6.92 | 34.03 |
| | 0.2 | | | 5.32 | 31.21 | 5.81 | 31.43 | $\mathbf{6.08}$ | $\mathbf{31.56}$ |

methods on test, with the exception of the WideResNet on CIFAR-10/100 and the PreAct-ResNet on Tiny-ImageNet. In combination with PyramidNet, FMix achieves, to the best of our knowledge, a new state-of-the-art classification accuracy on CIFAR-10 without use of external data. By the addition of Fast AutoAugment, this setting bares some similarity to the recently proposed AugMix (Hendrycks et al., 2019a) which performs MixUp on heavily augmented variants of the same image. With the PreAct-ResNet18, FMix obtains a new state-of-the-art classification accuracy on Fashion MNIST. Note that Zhang et al. (2017) also performed experiments with the PreAct-ResNet18, WideResNet-28-10, and DenseNet-BC-190 on CIFAR-10 and CIFAR-100. There are some discrepancies between the authors results and the results obtained by our implementation. Whether any differences are significant is difficult to ascertain as no measure of deviation is provided in Zhang et al. (2017). However, since our implementation is based on the implementation from Zhang et al. (2017), and most of the differences are small, we have no reason to doubt it. We speculate that these discrepancies are simply a result of random initialisation, but could also be due to differences in reporting or training configuration.

Next, we obtain classification results on the ImageNet Large Scale Visual Recognition Challenge (ILSVRC2012) data set (Russakovsky et al., 2015). We train a ResNet-101 on the full data set (ImageNet), additionally evaluating on ImageNet-a (Hendrycks et al., 2019b), a set of natural adversarial examples to ImageNet models, to determine adversarial robustness. We train for 90 epochs with a batch size of 256. We perform experiments with both $\alpha = 1.0$ and $\alpha = 0.2$ (as this was used by Zhang et al. (2017)). The results, given in Table 3, show that FMix was the only MSDA to provide an improvement over the baseline with these hyper-parameters. Note that MixUp obtains an accuracy of 78.5 in Zhang et al. (2017) when using a batch size of 1024. Additionally note that MixUp obtains an accuracy of 79.48 and CutMix obtains an accuracy of 79.83 in Yun et al. (2019) when training for 300 epochs. Due to hardware constraints we cannot replicate these settings and so it is

Table 4: Classification performance for FMix against baselines on Bengali grapheme classification.

| Category | Baseline | FMix | MixUp | CutMix |
|---|---|---|---|---|
| Root | $92.86_{\pm 0.20}$ | $\mathbf{96.13}_{\pm 0.14}$ | $94.80_{\pm 0.10}$ | $95.74_{\pm 0.20}$ |
| Consonant diacritic | $96.23_{\pm 0.35}$ | $\mathbf{97.05}_{\pm 0.23}$ | $96.42_{\pm 0.42}$ | $\mathbf{96.96}_{\pm 0.21}$ |
| Vowel diacritic | $96.91_{\pm 0.19}$ | $\mathbf{97.77}_{\pm 0.30}$ | $96.74_{\pm 0.95}$ | $97.37_{\pm 0.60}$ |
| Grapheme | $87.60_{\pm 0.45}$ | $\mathbf{91.87}_{\pm 0.30}$ | $89.23_{\pm 1.04}$ | $91.08_{\pm 0.49}$ |

Table 5: Classification performance of FMix and baselines on sentiment analysis tasks.

| Data set | Model | Baseline | FMix | MixUp |
|---|---|---|---|---|
| Toxic (ROC-AUC) | CNN | $96.04_{\pm 0.16}$ | $\mathbf{96.80}_{\pm 0.06}$ | $96.62_{\pm 0.10}$ |
| | BiLSTM | $96.72_{\pm 0.04}$ | $\mathbf{97.35}_{\pm 0.05}$ | $97.15_{\pm 0.06}$ |
| | Bert ($\alpha$=0.1) | $98.22_{\pm 0.03}$ | $\mathbf{98.26}_{\pm 0.03}$ | - |
| IMDb | CNN ($\alpha$=0.2) | $86.68_{\pm 0.50}$ | $87.31_{\pm 0.34}$ | $\mathbf{88.94}_{\pm 0.13}$ |
| | BiLSTM ($\alpha$=0.2) | $88.29_{\pm 0.17}$ | $88.47_{\pm 0.24}$ | $\mathbf{88.72}_{\pm 0.17}$ |
| Yelp Binary | CNN | $95.47_{\pm 0.08}$ | $95.80_{\pm 0.14}$ | $\mathbf{95.91}_{\pm 0.10}$ |
| | BiLSTM | $96.41_{\pm 0.05}$ | $96.68_{\pm 0.06}$ | $\mathbf{96.71}_{\pm 0.07}$ |
| Yelp Fine-grained | CNN | $63.78_{\pm 0.18}$ | $\mathbf{64.46}_{\pm 0.07}$ | $\mathbf{64.56}_{\pm 0.12}$ |
| | BiLSTM | $62.96_{\pm 0.18}$ | $\mathbf{66.46}_{\pm 0.13}$ | $66.11_{\pm 0.13}$ |

not known how FMix would compare. On ImageNet-a, the general finding is that MSDA gives a good improvement in robustness to adversarial examples. Interestingly, MixUp with $\alpha = 1.0$ yields a lower accuracy on ImageNet but a much higher accuracy on ImageNet-a, suggesting that models trained with MixUp learn a fundamentally different function.

For a final experiment with image data, we use the Bengali.AI handwritten grapheme classification data set (Bengali.AI, 2020), from a recent Kaggle competition. Classifying graphemes is a multi-class problem, they consist of a root graphical form (a vowel or consonant, 168 classes) which is modified by the addition of other vowel (11 classes) or consonant (7 classes) diacritics. To correctly classify the grapheme requires classifying each of these individually, where only the root is necessarily always present. We train separate models for each sub-class, and report the individual classification accuracies and the combined accuracy (where the output is considered correct only if all three predictions are correct). We report results for 5 folds where 80% of the data is used for training and the rest for testing. We extract the region of the image which contains the grapheme and resize to $64 \times 64$, performing no additional augmentation. The results for these experiments, with an SE-ResNeXt-50 (Xie et al., 2017; Hu et al., 2018), are given in Table 4. FMix and CutMix both clearly offer strong improvement over the baseline and MixUp, with FMix performing significantly better than CutMix on the root and vowel classification tasks. As a result, FMix obtains a significant improvement when classifying the whole grapheme. In addition, note that FMix was used in the competition by Singer and Gordeev (2020) in their second place prize-winning solution. This was the best result obtained with MSDA.

**Sentiment Analysis** Although typically restricted to classification of two dimensional data, we can extend the MSDA formulation for classification of one dimensional data. In Table 5, we perform a series of experiments with MSDAs for the purpose of sentiment analysis. In order for MSDA to be effective, we group elements into batches of similar sequence length as is already a standard practice. This ensures that the mixing does not introduce multiple end tokens or other strange artefacts (as would be the case if batches were padded to a fixed length). The models used are: pre-trained FastText-300d (Joulin et al., 2016) embedding followed by a simple three layer CNN (LeCun et al., 1995), the FastText embedding followed by a two layer bi-directional LSTM (Hochreiter and Schmidhuber, 1997), and pre-trained Bert (Devlin et al., 2018) provided by the HuggingFace transformers library (Wolf et al., 2019). For the LSTM and CNN models we compare MixUp and FMix with a baseline. For the Bert fine-tuning we do not compare to MixUp as the model input is a series of tokens, interpolations between which are meaningless. We first report results on the

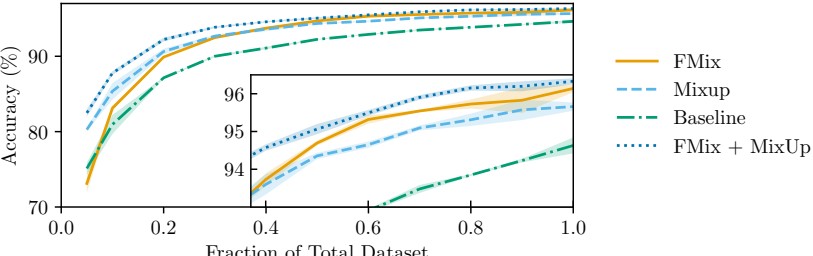

Figure 3: CIFAR-10 performance for a PreAct-ResNet18 as we change the amount of training data.

Toxic Comments (Jigsaw and Google, 2018) data set, a Kaggle competition to classify text into one of 6 classes. For this data set we report the ROC-AUC metric, as this was used in the competition. Note that these results are computed over the whole test set and are therefore not comparable to the competition scores, which were computed over a subset of the test data. In this setting, both MixUp and FMix provide an improvement over the baseline, with FMix consistently providing a further improvement over MixUp. The improvement when fine-tuning Bert with FMix is outside the margin of error of the baseline, but mild in comparison to the improvement obtained in the other settings. We additionally report results on the IMDb (Maas et al., 2011), Yelp binary, and Yelp fine-grained (Zhang et al., 2015) data sets. For the IMDb data set, which has one tenth of the number of examples, we found $\alpha = 0.2$ to give the best results for both MSDAs. Here, MixUp provides a clear improvement over both FMix and the baseline for both models. This suggests that MixUp may perform better when there are fewer examples.

**Combining MSDAs**   We have established through our analysis that models trained with interpolative MSDA perform a fundamentally different function to models trained with masking. We now wish to understand whether the benefits of interpolation and masking are mutually exclusive. We therefore performed experiments with simultaneous action of multiple MSDAs, alternating their application per batch with a PreAct-ResNet18 on CIFAR-10. A combination of interpolation and masking, particularly FMix+MixUp ($96.30_{\pm 0.08}$), gives the best results, with CutMix+MixUp performing slightly worse ($96.26_{\pm 0.04}$). In contrast, combining FMix and CutMix gives worse results ($95.85_{\pm 0.1}$) than using either method on its own. For a final experiment, we note that our results suggest that interpolation performs better when there is less data available (e.g. the IMDb data set) and that masking performs better when there is more data available (e.g. ImageNet and the Bengali.AI data set). This finding is supported by our analysis since it is always easier for the model to learn specific features, and so we would naturally expect that preventing this is of greater utility, when there is less data. We confirm this empirically by varying the size of the CIFAR-10 training set and training with different MSDAs in Figure 3. Notably, the FMix+MixUp policy obtains superior performance irrespective of the amount of available data.

## 6   CONCLUSIONS AND FUTURE WORK

In this paper we have introduced FMix, a masking MSDA that improves classification performance for a series of models, modalities, and dimensionalities. We believe the strength of masking methods resides in preserving local features and we improve upon existing approaches by increasing the number of possible mask shapes. We have verified this intuition through a novel information theoretic analysis. Our analysis shows that interpolation causes models to encode more general features, whereas masking causes models to encode the same information as when trained with the original data whilst eliminating memorisation. Our preliminary experiments suggest that combining interpolative and masking MSDA could improve performance further, although further work is needed to fully understand this phenomenon. Future work should also look to expand on the finding that masking MSDA works well in combination with Fast AutoAugment (Lim et al., 2019), perhaps by experimenting with similar methods like AutoAugment (Cubuk et al., 2019a) or RandAugment (Cubuk et al., 2019b). Finally, our early experiments resulted in several lines of enquiry that ultimately did not bare fruit, which we discuss further in Appendix F.

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

## A    ON THE IMPORTANCE OF TARGETS

The standard formulation for classification with MSDA weights the cross entropy losses computed with each of the true labels by the corresponding input mixing ratio. It could be suggested that by mixing the targets differently, one might obtain better results. However, there are key observations from prior art which give us cause to doubt this supposition; in particular, Liang et al. (2018) performed a number of experiments on the importance of the mixing ratio of the labels in MixUp. They concluded that when the targets are not mixed in the same proportion as the inputs the model can be regularised to the point of underfitting. However, despite this conclusion their results show only a mild performance change even in the extreme event that targets are mixed randomly, independent of the inputs. That doesn't mean that the target space is always insignificant. For example, we might care about how calibrated the outputs are. Calibration is the extent to which an output 'probability' corresponds to the *actual* probability of being correct. Clearly, this is a challenging property to evaluate since we have no notion of ground truth uncertainty in the data. Peterson et al. (2019) suggest using human uncertainty as a baseline on the CIFAR-10 data set. Specifically, they introduce the CIFAR-10H data set which consists of human soft-labels for the CIFAR-10 test set, i.e. the distribution resulting from many different humans labelling each image. We evaluate a series of CIFAR-10 pretrained PreAct-ResNet18 models on CIFAR-10H in Table 6. The metric used is the relative entropy of the model outputs with respect to the soft-labels. The results show that the masking MSDA approaches induce a notion of uncertainty that is more similar to that of human observers. An important weakness of this claim derives from the cross entropy objective used to train models. We note that

$$H(p_{\hat{Y} \mid X}, p_{Y \mid X}) = H(p_{\hat{Y} \mid X}) + D\big(p_{\hat{Y} \mid X} \,\|\, p_{Y \mid X}\big) \;. \tag{4}$$

In other words, the model is jointly required to match the target distribution and minimise the entropy of each output. The result of this is that trained models naturally output very high confidence predictions as an artefact of their training process. The above claim should therefore be taken with a pinch of salt since it is likely that the improved results derive simply from the lower entropy targets and model outputs. Furthermore, we expect that significant improvement would be gained in this test by training MSDA models with a relative entropy objective rather than the cross entropy.

Table 6: Mean and standard deviation divergence scores on CIFAR-10H, using the PreAct ResNet18 model trained on CIFAR-10.

|  | Baseline | FMix | MixUp | CutMix |
|---|---|---|---|---|
| $D\big(p_{\hat{Y} \mid X} \,\|\, p_{Y_H \mid X}\big)$ | $0.716_{\pm 0.032}$ | $0.220_{\pm 0.009}$ | $0.239_{\pm 0.005}$ | $\mathbf{0.211}_{\pm 0.005}$ |

## B    VAE MUTUAL INFORMATION

Recall from the paper that we wish to estimate the mutual information between the representation learned by a VAE from the original data set, $Z_X$, and the representation learned from some augmented data set, $Z_A$, written $I(Z_X; Z_A) = \mathbb{E}_{Z_X}\big[D\big(p_{(Z_A \mid Z_X)} \,\|\, p_{Z_A}\big)\big]$. VAEs comprise an encoder, $p_{(Z \mid X)}$, and a decoder, $p_{(X \mid Z)}$. We impose a standard Normal prior on $Z$, and train the model to maximise the Evidence Lower BOund (ELBO) objective

$$\mathcal{L} = \mathbb{E}_X\big[\mathbb{E}_{Z \mid X}\big[\log(p_{(X \mid Z)})\big] - D\big(p_{(Z \mid X)} \,\|\, \mathcal{N}(\mathbf{0}, I)\big)\big] \;. \tag{5}$$

Denoting the outputs of the decoder of the VAE trained on the augmentation as $\hat{X} = decode(Z_X)$, and by the data processing inequality, we have $I(Z_A; \hat{X}) \leq I(Z_A; Z_X)$ with equality when the decoder retains all of the information in $Z$. Now, we need only observe that we already have a model of $p_{(Z_A \mid X)}$, the encoder trained on the augmented data. Estimating the marginal $p_{Z_A}$ presents a challenge as it is a Gaussian mixture. However, we can measure an alternative form of the mutual information that is equivalent up to an additive constant, and for which the divergence has a closed form solution, with

$$\mathbb{E}_{\hat{X}}\big[D\big(p_{(Z_A \mid \hat{X})} \,\|\, p_{Z_A}\big)\big] = \mathbb{E}_{\hat{X}}\big[D\big(p_{(Z_A \mid \hat{X})} \,\|\, \mathcal{N}(\mathbf{0}, I)\big)\big] - D\big(p_{Z_A} \,\|\, \mathcal{N}(\mathbf{0}, I)\big) \;. \tag{6}$$

The above holds for any choice of distribution that does not depend on $\hat{X}$. Conceptually, this states that we will always lose more information on average if we approximate $p_{(Z_A \mid \hat{X})}$ with any constant distribution other than the marginal $p_{Z_A}$. Additionally note that we implicitly minimise $D\big(p_{Z_A} \,\|\, \mathcal{N}(\mathbf{0}, I)\big)$ during training of the VAE (Hoffman and Johnson, 2016). In light of this fact, we can write $I(Z_A; \hat{X}) \approx \mathbb{E}_{\hat{X}}[D\big(p_{(Z_A \mid \hat{X})} \,\|\, \mathcal{N}(\mathbf{0}, I)\big)]$.

We can now easily obtain a helpful upper bound of $I(Z_A; Z_X)$ such that it is bounded on both sides. Since $Z_A$ is just a function of $X$, again by the data processing inequality, we have $I(Z_A; X) \geq I(Z_A; Z_X)$. This is easy to compute since it is just the relative entropy term from the ELBO objective. To summarise, we can compute our measure by first training two VAEs, one on the original data and one on the augmented data. We then generate reconstructions of data points in the original data with one VAE and encode them in the other. We now compute the expected value of the relative entropy between the encoded distribution and an estimate of the marginal to obtain an estimate of a lower bound of the mutual information between the representations. We then recompute this using real data points instead of reconstructions to obtain an upper bound.

## C SUPPLEMENTARY ANALYSES

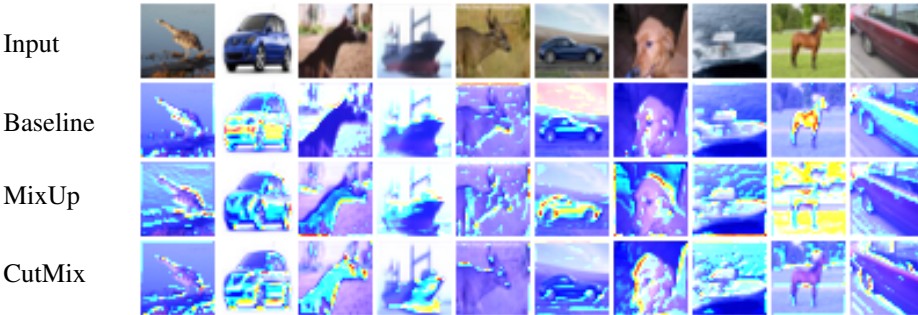

Figure 4: Grad-CAM at the output of the first convolutional layer of a PreAct-ResNet18 trained with a range of MSDAs.

## D ADDITIONAL EXPERIMENTS

Table 7: Classification performance for our approach, FMix, against a baseline for a PointNet (Qi et al., 2017) on ModelNet10 (Wu et al., 2015)

| Data set | Model | Baseline | FMix | MixUp | CutMix |
|---|---|---|---|---|---|
| ModelNet10 | PointNet | $89.10_{\pm 0.32}$ | $\mathbf{89.57}_{\pm 0.44}$ | - | - |
| Commands | ResNet ($\alpha{=}1.0$)
ResNet ($\alpha{=}0.2$) | $97.69_{\pm 0.04}$ | $\mathbf{98.59}_{\pm 0.03}$
$\mathbf{98.44}_{\pm 0.06}$ | $98.46_{\pm 0.08}$
$98.31_{\pm 0.08}$ | $98.46_{\pm 0.08}$
$\mathbf{98.48}_{\pm 0.06}$ |

**Point Cloud Classification**  We now demonstrate the extension of FMix to 3D through point cloud classification on ModelNet10 (Wu et al., 2015). We transform the pointclouds to a voxel representation before applying a 3D FMix mask. Table 7 reports the average median accuracy from the last 5 epochs, due to large variability in the results. It shows that FMix continues to improve results within significance, even in higher dimensions.

**Audio Classification**  The Google Commands data set was created to promote deep learning research on speech recognition problems. It is comprised of 65,000 one second utterances of one of 30 words, with 10 of those words being the target classes and the rest considered unrelated or background noise. We perform MSDA on a Mel-frequency spectrogram of each utterance. The

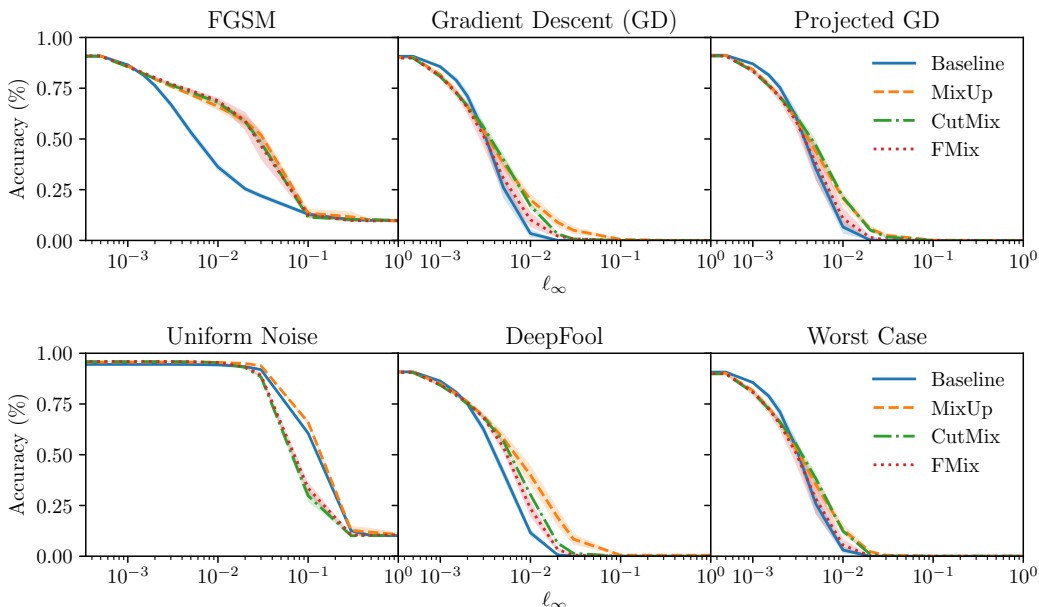

Figure 5: Full set of adversarial robustness results over the CIFAR-10 test set for a range of MSDAs, including our approach FMix.

results for a PreAct ResNet-18 are given in Table 7. We evaluate FMix, MixUp, and CutMix for the standard $\alpha = 1$ used for the majority of our experiments and $\alpha = 0.2$ recommended by Zhang et al. (2017) for MixUp. We see in both cases that FMix and CutMix improve performance over MixUp outside the margin of error, with the best result achieved by FMix with $\alpha = 1$.

# E    EXPERIMENTAL DETAILS

In this section we provide the experimental details for all experiments presented in the main paper. Unless otherwise stated, the following parameters are chosen: $\alpha = 1$, $\delta = 3$, weight decay of $1 \times 10^4$ and optimised using SGD with momentum of 0.9. For cross validation experiments, 3 or 5 folds of 10% of the training data are generated and used for a single run each. Test set experiments use the entire training set and give evaluations on the test sets provided. If no test set is provided then a constant validation set of 10% of the available data is used. Table 8 provides general training details that were present in all experiments.

All experiments were run on a single GTX1080ti or V100, with the exceptions of ImageNet experiments (4 × GTX1080ti) and DenseNet/PyramidNet experiments (2 × V100). ResNet18 and LSTM experiments ran within 2 hours in all instances, PointNet experiments ran within 10 hours, WideResNet/DenseNet experiments ran within 2.5 days and auto-augment experiments ran within 10 days.

For all image experiments we use standard augmentations to normalise the image to $[0, 1]$ and perform random crops and random horizontal flips. For the google commands experiment we used the transforms and augmentations implemented here `https://github.com/tugstugi/pytorch-speech-commands` for their solution to the tensorflow speech regonition challenge.

Table 8: General experimental details present in all experiments. Batch Size (BS), Learning Rate (LR). Schedule reports the epochs at which the learning rate was multiplied by 0.1. [†] Adam optimiser used.

| Experiment | Model | Epochs | Schedule | BS | LR |
|---|---|---|---|---|---|
| CIFAR-10 / 100 | PreAct-ResNet18 | 200 | 100, 150 | 128 | 0.1 |
| | WideResNet-28-10 | 200 | 100, 150 | 128 | 0.1 |
| | DenseNet-BC-190 | 300 | 100, 150, 225 | 32 | 0.1 |
| | PyramidNet-272-200 | 1800 | Cosine-Annealed | 64 | 0 - 0.05 |
| FashionMNIST | PreAct-ResNet18 | 200 | 100, 150 | 128 | 0.1 |
| | WideResNet-28-10 | 300 | 100, 150, 225 | 32 | 0.1 |
| | DenseNet-BC-190 | 300 | 100, 150, 225 | 32 | 0.1 |
| Google Commands | PreAct-ResNet18 | 90 | 30, 60, 80 | 128 | 0.1 |
| ImageNet | ResNet101 | 90 | 30, 60, 80 | 256 | 0.4 |
| TinyImageNet | PreAct-ResNet18 | 200 | 150, 180 | 128 | 0.1 |
| Bengali.AI | PreAct-ResNet18 | 100 | 50, 75 | 512 | 0.1 |
| Sentiment Analysis[†] | CNN | 15 | 10 | 64 | $10^{-3}$ |
| | LSTM | 15 | 10 | 64 | $10^{-3}$ |
| | Bert | 5 | 3 | 32 | $10^{-5}$ |
| Combining MSDAs | PreAct-ResNet18 | 200 | 100, 150 | 128 | 0.1 |
| ModelNet10[†] | PointNet | 50 | 10, 20, 30, 40 | 16 | $10^{-3}$ |
| Ablations | PreAct-ResNet18 | 200 | 100, 150 | 128 | 0.1 |

# F    THINGS WE TRIED THAT DIDN'T WORK

This section details a number of experiments and modifications we attempted which did not lead to significant results. Our aim here is to prevent future research effort being devoted to approaches that have already been explored by us. It may also be the case that better versions of these could be constructed which obtain better results.

## F.1    SALIENCE PRIOR

It is clear that we should care about how the mixing coefficient relates to the relative amount of salient information from each data point in the outcome. This presents a challenge because getting $\lambda$ of the salient information in the first data point does not imply that we have $1 - \lambda$ of the salient information in the second. We could consider making an assumption that the expected distribution of salient information in each data point is the same. In such a case, the above problem no longer exists. For images, a simple assumption would be that the salient information is roughly Gaussian about the centre. To apply a salience prior to our mask generation process, we need to change the binarisation algorithm. Specifically, we iterate over the values in descending order until the mass over the prior is equal to $\lambda$. We experimented with this approach and found no significant performance gain, and so did not pursue it any further. That said, there may still be some value to the above motivation and a more complex, data point specific, salience distribution could work.

## F.2    MASK SOFTENING

Following the observation that combining interpolation and masking provides the best results, and particularly the experiments in Summers and Dinneen (2019), we considered a grey-scale version of FMix. Specifically, we explored a method which softened the edges in the mask. To achieve this, after sorting the low frequency image by pixel value, instead of choosing a threshold and setting one side to 1 and the other to 0, we choose an equal distance either side of the threshold and linearly value the mask between 1 and 0 for some number of pixels. The number of grey pixels is chosen to ensure that the mean mask value is retained and that the fraction of the image that is non-binary does not exceed some present value.

We found that softening the masks resulted in no performance gains, and in fact, occasionally hindered training. We considered it again for the toxic comments experiments since we assumed smooth transitions would be very important for text models. It did offer minor improvements over default FMix, however, we judged that the gain was not worth the added complexity and diluting of the core idea of FMix for us to present it in the paper. Furthermore, proposing it for the singular case of toxic comments would have been bad practice, since we only observed an improvement for one model, on one data set. That said, we feel mask softening would be interesting to explore further, certainly in the case of text models. We would need to experiment with softened FMix masks in multiple text data sets and observe improvement in most or all of them over base FMix in order to formally propose softening as an FMix modification.

## F.3    TARGET DISTRIBUTION

A final alteration that we experimented with relates to the distribution of targets. The idea was that we could change the distribution of the target mixing coefficients to obtain better 'calibrated' model outputs. The way this is done is simple, we pass the sampled $\lambda$ through its CDF and then through the inverse CDF of the target distribution. This allows us to, for example, encourage confident outputs by choosing a symmetric Beta distribution with $\alpha \approx 0.1$. The issue with this approach is two fold. First, changing the distribution of the outputs in this way has no bearing on the ordering, and so no effect on the classification accuracy. Second, any simple transform of this nature can be trivially learned by the model or applied in post. In other words, it is equivalent to training a model normally and then just transforming the outputs. As a result, it is difficult to argue that this approach does anything particularly clever. We trained models with different target distributions at several points and found that the performance was not significantly different.

## G    CURRENT UNDERSTANDING OF MSDA

Attempts to explain the success of MSDAs were not only made when they were introduced, but also through subsequent empirical and theoretical studies. In this section we review these studies to paint a picture of the current theories, and points of contention, on how MSDA works. In addition to their experimentation with the targets, Liang et al. (2018) argue that linear interpolation of inputs limits the memorisation ability of the network. Gontijo-Lopes et al. (2020) proposes two measures to explain the impact of augmentation on generalisation when jointly optimised: affinity and diversity. While the former captures the shift in the data distribution as perceived by the baseline model, the latter measures the training loss when learning with augmented data. A somewhat more mathematical view on MSDA was adopted by Guo et al. (2019), who argue that MixUp regularises the model by constraining it outside the data manifold. They point out that this could lead to reducing the space of possible hypotheses, but could also lead to generated examples contradicting original ones, degrading quality. Upon Taylor-expanding the objective, a more recent study that also focuses on MixUp motivates its success by the co-action of four different regularisation factors (Carratino et al., 2020).

Following Zhang et al. (2017), He et al. (2019) take a statistical learning view of MSDA, basing their study on the observation that MSDA distorts the data distribution and thus does not perform VRM in the traditional sense. They subsequently propose separating features into 'minor' and 'major', where a feature is referred to as 'minor' if it is highly sample-specific. Augmentations that significantly affect the distribution are said to make the model predominantly learn from 'major' features. From an information theoretic perspective, ignoring these 'minor' features corresponds to increased compression of the input by the model. Although He et al. (2019) noted the importance of characterising the effect of data augmentation from an information perspective, they did not explore any measures that do so. Instead, He et al. (2019) analysed the variance in the learned representations. It can be seen that this is analogous to the entropy of the representation since entropy can be estimated via the pairwise distances between samples, with higher distances corresponding to both greater entropy and variance (Kolchinsky and Tracey, 2017). In proposing Manifold MixUp, Verma et al. (2019) additionally suggest that MixUp works by increasing compression. The authors compute the singular values of the representations in early layers of trained networks, with smaller singular values again corresponding to lower entropy. The issue with these approaches is that the entropy of the representation is only an upper bound on the information that the representation has about the input.

An issue with these findings is that they relate purely to interpolative MSDAs. It is also the case that there is disagreement in the conclusions of some of these studies. If interpolative MSDA works by preventing the model from learning about so called 'minor' features, then that would suggest that the underlying data distribution has been distorted, breaking the core assumption of VRM. Furthermore, Yun et al. (2019) suggested that masking MSDA approaches work by addressing this distortion. If this is the case then we should expect them to perform worse than interpolative MSDAs since the bias towards compressed representations has been removed. Clearly, there is some contention about the underlying mechanisms driving generalisation in MSDAs. In particular, it is necessary to provide an explanation for masking MSDAs that is complementary to the current explanations of interpolative MSDAs, rather than contradictory to them.

## H  HYPERPARAMETER CHOICE

Figure 6a gives the relationship between validation accuracy and the parameter $\alpha$ for three MSDA methods. Validation accuracy is the average over 5 folds with a validation set consisting of $10\%$ of the data. This ablation was performed on the CIFAR-10 data set using the PreAct ResNet18 model from the previous experiments. In the cases of FMix and MixUp there exists an optimal value. In both cases, this point is close to $\alpha = 1$, although for MixUp it is skewed slightly toward 0, as was found for their ImageNet experiments. The choice of decay power $\delta$ is certainly more significant. Figure 6b shows that low values of $\delta$ drastically reduce the final accuracy. This is unsurprising since low $\delta$ corresponds to a speckled mask, with no large regions of either data point present in the augmentation. Larger values of $\delta$ correspond to smoother marks with large cohesive regions from each donor image. We note that for $\delta \gtrsim 3$ there is little improvement to be gained, validating our decision to use $\delta = 3$.

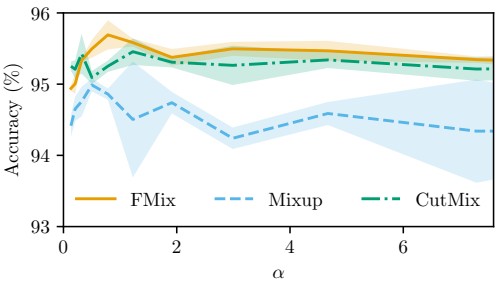
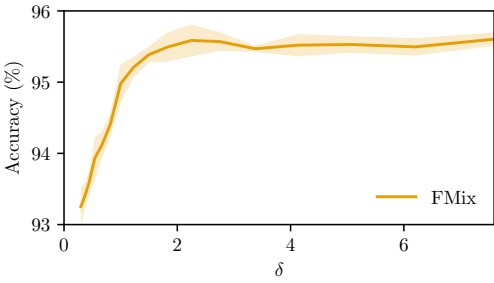

(a) Performance of masking MSDAs (FMix and Cut-Mix) remains with increased mixing (as $\alpha$ increases). Performance of interpolative MSDAs (MixUp) does degrade, since data level distortion increases.

(b) Performance of FMix increases with the decay power $\delta$. Using a lower frequency grey-scale image (increasing $\delta$) increases local consistency up to a point ($\delta \approx 3$).

Figure 6: CIFAR-10 accuracy for a PreAct-ResNet18 with varying $\alpha$ trained with FMix (ours), MixUp and CutMix (Figure 6a), and with varying $\delta$ trained with FMix (Figure 6b).

