# OpenReview forum: "FMix: Enhancing Mixed Sample Data Augmentation"
_ICLR.cc/2021/Conference — Reject_

### Official Review · AnonReviewer4 · 2020-10-27
**Good paper, but concerns in the conclusions from the analyses**

**Rating:** 6
**Confidence:** 4

**Review:**

The paper presents an interesting analysis of CutMix and MixUp data augmentation techniques. It also presents an improvement to CutMix that removes the horizontal/vertical axis bias. The idea to use fourier noise to construct masks for a variant of CutMix is interesting and well-motivated.

My main concern is whether the conclusions drawn by the analyses are fully grounded. The paper performs an analysis of the effect of the augmented data on learned representations by training unsupervised models on the augmented or clean data and measuring their mutual information. This analysis has the undesirable property of not matching the supervised case in a number of ways, such as different learning objectives, model architectures, etc. Even ignoring this, if we take the result that “MixUp consistently reduces the amount of information that is learned about the original data”, what then explains the improved generalization accuracy MixUp showcases in their original paper?

Moreover, after claiming that the analysis indicates that MixUp learns different representations, the paper asks “whether these different representations learned from MixUp give rise to practical differences other than just improved generalisation.” The issue is that they do this via an adversarial attack analysis, rather than a more realistic non-worst-case robustness analysis. This leads to the conclusion “MixUp (...) does not correspond to a general increase in robustness.” But it does not answer the original question of whether “MixUp gives rise to practical differences other than just improved generalisation.” The finding that MixUp yields greater ImageNet-A robustness (presented later in the paper) also contradicts this early claim.

The finding that MixUp provides more compressed representations does not necessarily mean that masking augmentation methods are better than interpolation ones. The paper seems to acknowledge this as well, in the final paragraph of the introduction, where it describes an experiment in which combining FMix+MixUp gives the best results (presumably because their representations of data are different and therefore combining them would yield the best of both worlds). This seems to contradict the previous adversarial analysis in which MixUp was found to not yield significantly more robustness. Further, the combination experiment has the two leading combination methods (FMix+MixUp and CutMix+MixUp) yield very similar results (within the margin of error), which opens the question of whether FMix meaningfully improves over CutMix.

Overall, I find the paper very easy to read and presenting some interesting ideas and even some exciting improvements in performance. I just wish the presentation and the claims made in the analysis of MSDA methods accounted for some of the inconsistencies described above.

Update after rebuttal: I appreciate the authors' response and clarifications. I maintain my original score.

---

> ### Author Response · Authors · 2020-11-17
> **Thank you for your review**
>
> We thank the reviewer for their comments. Our responses are as follows:
>
> __“what then explains the improved generalization accuracy MixUp showcases in their original paper [given that we claim MixUp reduces the amount of information that is learned about the original data]”__
> We take a reduction in the amount of information to indicate more compressed representations, which, from an information theoretic point of view explains the improvement in generalization.
>
> __“[the adversarial attack experiment] does not answer the original question of whether MixUp gives rise to practical differences other than just improved generalisation.”__
> The worst-case analysis only serves to summarize the results from the individual attacks. The individual results do illuminate differences between the trained models and particularly that MixUp trained models are the most different from the baseline (e.g. in robustness to DeepFool). That said, we will tone down the language here to make this clearer. We can move the robustness experiments to the appendix if the reviewer feels it would help. We would like to thank the reviewer for helping us create a more concise and stronger argument for our approach.
>
> __“The finding that MixUp yields greater ImageNet-A robustness (presented later in the paper) also contradicts this early claim.”__
> Since the ImageNet-A data is generated for a particular model (ImageNet trained ResNet-50 w/o MSDA) the ‘increased ImageNet-A robustness’ is perhaps better described as the MixUp model making mistakes in a different way to a baseline model. To determine robustness, we would need to re-generate the ImageNet-A set from each model (in which case we would expect to see all of the model's performance reduced to near zero). We will improve the writing here to make this more clear.
>
> __“[the paper] describes an experiment in which combining FMix+MixUp gives the best results (presumably because their representations of data are different and therefore combining them would yield the best of both worlds). This seems to contradict the previous adversarial analysis in which MixUp was found to not yield significantly more robustness.”__
> The point of the experiment where we combine masking and interpolation is simply to show that the differences between interpolation and masking can be jointly exploited. This does not contradict the robustness experiments since they make no claim about the generalization performance of models trained with the different methods.
>
> __“Further, the combination experiment has the two leading combination methods (FMix+MixUp and CutMix+MixUp) yield very similar results (within the margin of error), which opens the question of whether FMix meaningfully improves over CutMix.”__
> We would like to stress that we did not experiment with multiple ways of combining MSDA that may or may not lead to different results. As such, although this is a valid observation, many more experiments would be needed to properly assess the various possible combinations of different MSDAs and we are reluctant to make any strong claims regarding the results from this section.

---

### Official Review · AnonReviewer1 · 2020-10-28
**Weak logical connection between the motivation and the method, too small performance gap**

**Rating:** 4
**Confidence:** 5

**Review:**

This paper proposes an advanced masking strategy for CutMix augmentation based on the low-pass filter. The authors provide an interesting mutual information analysis for different augmentation strategies to describe their motivation. The experiments include many vision tasks (CIFAR-10, CIFAR-100, Fashion-MNIST, Tiny-ImageNet, ImageNet, Bengali datasets) and language tasks (Toxic, IMDb, Yelp).

**Pros**

\+ The mutual information analysis provides us a new perspective to understand different data augmentations.

\+ Various experiments.

**Cons**

**[Contradictory results between mutual information and performances]**
If we believe the VAE experiments in section 3, we have another paradox: the mutual information measurement and the real performance are not related.
Table 1 shows that in terms of mutual information, MixUp < Baseline < CutMix (and < FMix with a very small gap).
However, many experiments in this paper show that baseline < mixup < cutmix in terms of the performances.
This paper cites information bottleneck theory to justify the deceases shown by Mixup, but it is still contradictory to the performances.
It makes me confused to understand the meaning of mutual information. What is good for an augmentation method if we have high or low mutual information? It is still unclear to me.

A similar comment also can be applicable to the "adversarial robustness" experiments. Aside from that mixup is hard to say "adversarial training" (what is the threat model in this scenario?), I feel that this result is irrelevant to FMix motivation.


**[Weak logical connection between the motivation and the method]**
In my opinion, the connection between the analysis in the motivation and the proposed method is too weak. This paper proposes a CutMix variant where the mask is sampled by a low-pass filter. Why the low-pass filter approach can solve the motivation, i.e., enhancing mutual information between input and augmented images? There could be other possible variants as discussed in my "related works" comment


**[Related works]**
There are a few CutMix variants that employ a non-random masking strategy. Especially, I believe these two variants, which have similar motivation, should be compared:

- Walawalkar, Devesh, et al. "Attentive Cutmix: An Enhanced Data Augmentation Approach for Deep Learning Based Image Classification." ICASSP 2020
- Kim, Jang-Hyun, Wonho Choo, and Hyun Oh Song. "Puzzle mix: Exploiting saliency and local statistics for optimal mixup." ICML 2020

where Attentive CutMix uses CAM to extract masks, and PuzzleMix employs an optimization problem to optimize masks.
If it is possible, please provide more comparison between these two papers.


**[Too small performance gap, less convincing experiments]**
In Table 2, the performance gap between FMix and CutMix is too small, usually less than 0.3%. Note that the performance gaps are almost neglectable in these tasks.

Furthermore, FMix is often worse than CutMix in many tasks (Table 2 TinyImageNet, Table 3 ImageNet-A, Table 4, CIFAR-10H Table 6). I wonder what is the advantage to use FMix comparing to CutMix if FMix shows worse performance than CutMix.

Especially, I believe Table 3 is problematic. This paper argues that "Mixup uses 1024 batch size and CutMix uses 300 epochs". However, in PuzzleMix Table 5, CutMix-trained ResNet50 (top1 err 22.92) outperforms baseline ResNet50 (top1 err 24.31) with only 100 epochs. Thus, to me, this table is not convincing enough.


**[Potential issues in VAE analysis]**
The mutual information analysis is heavily relying on the learned VAE model. I wonder the quality of the generated images by VAE, in terms of both qualitatively (please provide generated samples in the supplementary) and quantitatively (e.g., FID).
If the VAE is not optimized well, the analysis will not be convincing enough.


**Minor comments**
- Why CutMix experiments are missed in Table 5?
- I suggest avoiding using the words, "clear" and "clearly".

---

**Post-rebuttal update**

My main concerns in the initial review were three-folds:

- Potential flaws in the analyses based on VAE and adversarial attacks
- Unclear connection between the MI analysis and the proposed method
- Small performance gap, and even sometimes worse performance, compared to the baseline methods (Mixup, CutMix)

After having discussions with the authors, I will keep my initial score because:

- I am still confused about the MI-based analysis conclusion. The authors mentioned *"We make no claim that increasing or decreasing the mutual information measure will have a strong impact on performance. Instead, we contend that MixUp works by forcing the model to ignore sample specific features (thus learning compressed representations – the reason for discussing the information bottleneck theory) and that CutMix works by mimicking the real data whilst preventing example memorization."* in the rebuttal, but these two conclusions are not trivial to me (by the MI analysis).
- Even if we ignore the first part, my second concern still remains. The authors mentioned *"That is the problem FMix tries to solve by removing the horizontal and vertical edge artefacts from cutmix. Our belief is that cutmix biases models towards these edges as they are a guaranteed feature of the data and learning about them would reduce the loss since these edges can tell you how much of each source image is present in the input (a key part of the objective)."*. But if this paper assumes that the rectangle masking strategy of CutMix makes bias, then I think other CutMix variants such as AttentiveCutMix or PuzzleMix should be considered as the comparison methods. Hence, I disagree with this statement *"A comparison to masks generated using additional models (and, thus, significant additional computation) does not seem fair to us."*
- For my last concern, the small performance gap, the authors claimed that this method *"was also used by the second place team in the BengaliAI Kaggle competition"*. It is good evidence that FMix can sometimes offer benefit to real-world applications, but I think more evidence that FMix can really solve problems of previous MSDA in a certain scenario, e.g., the edge bias as pointed by the authors.

---

> ### Author Response · Authors · 2020-11-17
> **Thank you for your review**
>
> We thank the reviewer for their comments. Our responses are as follows:
> __[Contradictory results between mutual information and performances]__
>
> We would like to clarify that the mutual information was not intended or expected to directly correlate with generalization performance. Our intention was merely to explore how learned representations are influenced by the different forms of MSDA. We make no claim that increasing or decreasing the mutual information measure will have a strong impact on performance. Instead, we contend that MixUp works by forcing the model to ignore sample specific features (thus learning compressed representations – the reason for discussing the information bottleneck theory) and that CutMix works by mimicking the real data whilst preventing example memorization. The result for FMix only serves to validate that FMix does a good job of mimicking the data, not as indicator of performance.
>
> Our intention with the adversarial robustness experiments was to show how these differences between learned functions (evidenced with our MI experiments) correspond to practical differences in how the models are impacted by out of distribution data (adversarial examples).
>
> __[Weak logical connection between the motivation and the method]__
> Unfortunately, we believe the reviewer misunderstood both our approach and our motivation. The masks in FMix are not sampled by a low-pass filter of a particular image. Instead, we sample masks __randomly__ from Fourier space. Figure 1 in the paper shows examples of the masks we use.
>
> Regarding the motivation, we would understand the reviewer’s concerns had we indeed claimed that our purpose was “enhancing mutual information between input and augmented images”. However, we do not claim this at any point in the paper. We state: “The results show that MixUp consistently reduces the amount of information that is learned about the original data. In contrast, CutMix manages to induce greater mutual information with the data than is obtained from just training on the un-augmented data”. Our analysis, as explained in the introduction of Section 5, concerns the learned representations and not the augmented images. We will reiterate this in the last part of this section to avoid future confusions. We hope this clarifies our approach and the connection between the motivation and the method.
>
> __[Related works]__
> A comparison to masks generated using additional models (and, thus, significant additional computation) does not seem fair to us. The purpose of our proposed augmentation is to increase performance with little to no additional impairments and we choose to compare to other methods that do so.
>
> __[Too small performance gap]__
> We appreciate that not all results show dramatic improvement over the alternatives, however, this could also be said for both CutMix and MixUp. Whilst FMix may not improve performance across the board, we believe it is better to have the option rather than not. Additionally, we believe that significance is best reflected in the impact on the community. FMix has been the starting point for further publications (e.g. FMixCutMatch [1]) and was also used by the second place team in the BengaliAI Kaggle competition. Please note that all of this was done independently of us.
>
> PuzzleMix doesn’t report the batch size used and performs other modifications to the training procedure. Such as resizing images differently depending on the epoch and learning rate jumps beyond cosine annealing with warm start. These methods were explicitly chosen [https://arxiv.org/pdf/2001.03994.pdf] to speed up ImageNet training, which whilst useful, is not the same as reporting performance on the base task. Furthermore, the resulting improvement over CutMix is under 0.5% despite supervised optimization of masks...
>
> __[Potential issues in VAE analysis]__
> We disagree with the statement that “If the VAE is not optimized well, the analysis will not be convincing enough.”. For any VAE our method allows for a comparison to be made. Although the values will likely change with different architectures the ordering should not - it would be wrong to assert that a more optimized VAE would necessarily be more informative. One interesting potential direction would be to vary the architecture to understand how these values change with weaker / stronger models. That said, we will include VAE generated samples and performance metrics in an appendix.
>
> __[Minor Comments]__
> CutMix is explicitly defined as a 2D method. Sentiment analysis augmentation is a 1D task.
>
> [1] Wei, X., Wei, X., Kong, X., Lu, S., Xing, W., & Lu, W. (2020). FMixCutMatch for semi-supervised deep learning. Neural Networks.

---

> > ### Comment · AnonReviewer1 · 2020-11-24
> > **Thanks for your answers**
> >
> > I thank the authors for answering my questions.
> > After carefully reading the rebuttal and other reviews, I think my main concern still remains.
> >
> > First, I'm still confusing how MI / adversarial analyses and the proposed random mask from Fourier space (btw, thanks for correcting my mistake). What does the MI analysis reveal? How the proposed method can solve or improve the proposed analyses?
> >
> > In the rebuttal, the authors claimed:
> >
> > > However, we do not claim this at any point in the paper. We state: “The results show that MixUp consistently reduces the amount of information that is learned about the original data. In contrast, CutMix manages to induce greater mutual information with the data than is obtained from just training on the un-augmented data”.
> >
> > To me, it is still a confusing argument. How FMix can solve the problems in theory or in conceptually? If the authors can answer this question before the final deadline (sorry for my late), it will be very delightful to me.
> >
> >
> > Furthermore, as my first review,  "I wonder what is the advantage to use FMix comparing to CutMix / Mixup if FMix shows worse performance than CutMix / Mixup."
> >
> > I don't think performance is everything but in this case, I'd expect one of strong theoretical support, conceptual inspiration, or significant performance gap.

---

> > > ### Author Response · Authors · 2020-11-24
> > > **Thank you for your reply**
> > >
> > > Thank you for your comments and for acknowledging the mistake in the initial review. We would like to take this opportunity to try to explain the story of how the paper came about and hopefully provide some insight into why we feel our work is of value.
> > >
> > > Our first motivations were to try to understand how mixup manages to work so well despite the mixed images not seeming to truly resemble the data. Furthermore, we were confused as to how cutmix can have the same effect despite the cutmixed images looking so different from the mixup images. This is what we feel our MI analysis starts to expand on. Actually, cutmix and mixup do very different things to the models trained with them but __both__ of those things can improve performance. Mixup prevents the model from learning about specific features (hence the lower information between the input and latent space) whereas cutmix simulates learning from the real data whilst preventing memorisation (more like a traditional augmentation such as random flipping would).
> > >
> > > __How FMix can solve the problems in theory or in conceptually?__
> > >
> > > Given this new understanding, we wondered whether there was some way to improve cutmix to make the augmented images resemble the real data even more. That is the problem FMix tries to solve by removing the horizontal and vertical edge artefacts from cutmix. Our belief is that cutmix biases models towards these edges as they are a guaranteed feature of the data and learning about them would reduce the loss since these edges can tell you how much of each source image is present in the input (a key part of the objective). In contrast, the edges in FMix are so inconsistent that learning about them is a much harder task and so the network is forced to learn about the actual data, rather than the augmentation.
> > >
> > > __Performance__
> > >
> > > Regarding performance our above understanding tells us that each of the MSDAs will be advantageous in different settings. This is because whether you would rather avoid specific features or avoid memorisation will depend heavily on the data set, model etc. Although FMix does not always provide a significant performance boost over the alternatives it is an important option that is sometimes the difference (as with the Bengali competition) between mediocre performance and prize winning performance. Ultimitely, we have struggled with comments regarding performance as we have tried to be as unbiased as possible in our presentation of the results by reporting all of the experiments we performed and not aggresively tuning our method over the others. If we had omitted the cases where our method didn't win and devoted a lot of resources to finding the best FMix parameters in every setting we no doubt would have seen more impressive numbers but at the unacceptable (to us) cost of no longer giving an honest reflection of the real world performance of FMix.

---

### Official Review · AnonReviewer3 · 2020-10-28
**Review: FMix: Enhancing Mixed Sample Data Augmentation**

**Rating:** 6
**Confidence:** 3

**Review:**

This paper introduces a new mixup method that builds masks by first sampling a grey-scale mask from fourier space, which is subsequently transformed into a binary mask. This improves results against several baselines and achieves state-of-the-art on a few important vision benchmarks.

My first remark is about the masking. The procedure seems fine, but why not compare to the way masks are sampled in context encoders [1]. This seems like an important baseline masking method to compare to. In addition, one could try sampling masks from a standard segmentation model, e.g., R-CNN.

My second remark is about the MI bounds. On page 14 in the Appendix, you state that the MI between Z_A and X_hat is approximately equal to the KL divergence between the posterior and the normal distribution, but in general this wont be true as in training the Gaussian mixture p_Za wont match the normal distribution, so you have an upper bound. So you have a lower bound of an upper bound to the MI, not a lower bound.

I'm curious though why not just use one of the recent neural estimators, e.g., found in [2] or [3]. In general, using VAEs for MI estimators depends heavily on the quality of the generator, so these neural estimators might be better suited.

Other comments:
P1:
* "'post-processing cannot increase information'" if such processing is deterministic, no?

P2:
* "CutMix imposes an unnecessary limitation": what limitation? Could you clarify?

Finally, do you plan to have updated results that compare to the 1024 batch size / 300 epoch settings?

[1] Pathak, Deepak, et al. "Context encoders: Feature learning by inpainting." Proceedings of the IEEE conference on computer vision and pattern recognition. 2016.
[2] Belghazi, Mohamed Ishmael, et al. "Mine: mutual information neural estimation." arXiv preprint arXiv:1801.04062 (2018).
[3] Poole, Ben, et al. "On variational bounds of mutual information." arXiv preprint arXiv:1905.06922 (2019).

---

> ### Author Response · Authors · 2020-11-17
> **Thank you for your review**
>
> We thank the reviewer for their comments. Our responses are as follows (numbered in order of appearance):
> 1. Both suggested approaches would likely provide some improvement. However, the purpose of our proposed augmentation is to increase performance with little to no additional resources or computation (the approach from context encoders would require additional loading and manipulation of mask images). As such, we only compare to other methods for which this is true. That said, these are valid suggestions, we will add them to the future work.
> 2. It is correct to say that our approximation of the MI is in fact an upper bound. However, the divergence between the marginal $p_{Z_A}$ and the prior is implicitly minimized during training. Furthermore, there is no reason to think that this value should deviate greatly between independent runs since this part of the objective can be satisfied arbitrarily during training without hurting any other (it is always possible to translate / scale the conditional distributions such that the marginal more closely fits the prior without losing any information about the input). We did consider an alternative which would use an unbiased sample from the marginal obtained by independently sampling an additional data point and passing that through the model (an approach used by InfoVAE). Please let us know if you would find this to be more convincing.
> 3. Some of our early experiments did use MINE, however, we found that it was prone to instability issues and getting reliable results that were consistent across runs was virtually impossible. We will add some discussion of these early investigations to the paper.
> 4. Regarding the point about generator quality, it is true that a more powerful VAE would retain more information about the data. However, our primary concern was with assessing the learned representations rather than the images themselves. For any fixed model architecture, independent of its performance we can obtain values which permit relative comparisons. One interesting potential direction would be to vary the architecture to understand how these values change with weaker / stronger models.
> 5. This should say “post-processing cannot increase information about the input”. Non-deterministic post-processing can increase entropy but not the mutual information about the input (e.g. given an X, no function X -> Y can yield a value which has more information about X than X does).
> 6. As mentioned in the beginning of the phrase, CutMix is limited to using square masks only. As evidenced in our paper, using a larger variety of masks can improve performance.
> 7. We do not plan to run additional experiments with such a resource-intensive setting as it is unclear what the scientific gain is in doing so (we would argue that ImageNet is a bad test of an augmentation approach in many ways since it is already pre-augmented with $\approx 1.2$ million examples). We focused on providing experiments for a wide range of applications and data sets and we believe this is more valuable than having a limited pool of experiments with even larger batch sizes and more epochs.

---

### Official Review · AnonReviewer2 · 2020-10-29
**A new variant of cutmix but the improvement is marginal**

**Rating:** 5
**Confidence:** 4

**Review:**

In this work, authors provide an analysis of mutual information for MSDA and the develop a new variant of mixup. The effectiveness is demonstrated by experiments.

Strength
1.	Authors study the difference between masking MSDA and interpolative MSDA, which is helpful for understanding the power of mixup and its variants.
2.	They develop a new augmentation method and improve the performance of masking MSDA.

Weakness
1.	The proposed measurement is not helpful for designing new methods. Note that the mutual information in mixup is lower than baseline while mixup still outperforms baseline.
2.	Compared to mixup and cutmix, the improvement reported in Table 2 is marginal.
3.	The experiments on ImageNet is unconvincing. Both of mixup and cutmix are worse than baseline, which contradicts the existing results.
4.	There lacks the discussion for the saliency based mixup methods, e.g., Puzzle Mix [1]. It is closely related to fmix but equipped with a learnable strategy to obtain patches for mixing.

[1] J-H Kim, et al. Puzzle mix: Exploiting saliency and local statistics for optimal mixup

---

> ### Author Response · Authors · 2020-11-17
> **Thank you for your review**
>
> We thank the reviewer for their comments. Our responses to the stated weaknesses are as follows:
> 1. We would like to clarify that the mutual information was not intended or expected to directly correlate with generalization performance. Our intention was merely to explore how learned representations are influenced by the different forms of MSDA. We make no claim that increasing or decreasing the mutual information measure will have a strong impact on performance. Instead, we contend that MixUp works by forcing the model to ignore sample specific features and that CutMix works by mimicking the real data whilst preventing example memorization. The result for FMix only serves to validate that FMix does a good job of mimicking the data, not as indicator of performance. We would welcome any suggestions on how this could be made clearer in our work.
> 2. We appreciate that not all results show dramatic improvement over the alternatives, however, this could also be said for both CutMix and MixUp. Whilst FMix may not improve performance across the board, we believe it is better to have the option rather than not. Additionally, we believe that significance is best reflected in the impact on the community. FMix has been the starting point for further publications (e.g. FMixCutMatch [1]) and was also used by the second place team in the BengaliAI Kaggle competition. Please note that all of this was done independently of us.
> 3. As mentioned in the paper, ImageNet performance is heavily dependent on hyperparameter choices. The results we provide were obtained with different hyperparameters than those used in the MixUp / CutMix papers due to computational restrictions. Importantly, our results __do not__ contradict the results from the respective works – it is simply the case that training with different hyperparameters will yield different results. The only alternative would be to not include the ImageNet results. However, we feel it is important to report all results obtained regardless of their potential for controversy.
> 4. We will reference saliency-based augmentation methods in our related work section. However, we reserve the discussion and comparison to other mixed sample augmentations that are similar in both approach and computational requirements (PuzzleMix requires costly additional computation that limits its value in practice).
>
> [1] Wei, X., Wei, X., Kong, X., Lu, S., Xing, W., & Lu, W. (2020). FMixCutMatch for semi-supervised deep learning. Neural Networks.

---

### Decision · Program_Chairs · 2021-01-07
**Final Decision**

**Decision:**

Reject

**Comment:**

The paper analyzes the space of mixed sample data augmentation approaches, and proposes a new variant, FMix, based on a new masking strategy. Reviewers point to the fact that FMix is only marginally better than previous approaches, that the experimental setup is unconvincing, and that the proposed analysis might not be grounded. This is a really borderline paper but I see the issues as more important than the benefits, so I recommend rejection.